# Increased flexibility of the SARS-CoV-2 RNA-binding site causes resistance to remdesivir

Shiho Torii[1☉¤a], Kwang Su Kim[2☉¤b], Jun Koseki[3☉], Rigel Suzuki[4☉], Shoya Iwanami[2], Yasuhisa Fujita[2], Yong Dam Jeong[2], Jumpei Ito[5], Hiroyuki Asakura[6], Mami Nagashima[6], Kenji Sadamasu[6], Kazuhisa Yoshimura[6], The Genotype to Phenotype Japan (G2P-Japan) Consortium[¶], Kei Sato[5,7,8,9,10,11], Yoshiharu Matsuura[1,12]*, Teppei Shimamura[3]*, Shingo Iwami[2,13,14,15,16,17]*, Takasuke Fukuhara[4,18]*

1 Laboratory of Virus Control, Research Institute for Microbial Diseases, Osaka University, Suita, Japan, 2 interdisciplinary Biology Laboratory (iBLab), Division of Biological Science, Graduate School of Science, Nagoya University, Nagoya, Japan, 3 Division of Systems Biology, Nagoya University Graduate School of Medicine, Nagoya, Japan, 4 Department of Microbiology and Immunology, Graduate School of Medicine, Hokkaido University, Sapporo, Japan, 5 Division of Systems Virology, Department of Microbiology and Immunology, The Institute of Medical Science, The University of Tokyo, Tokyo, Japan, 6 Tokyo Metropolitan Institute of Public Health, Tokyo, Japan, 7 Graduate School of Medicine, The University of Tokyo, Tokyo, Japan, 8 International Vaccine Design Center, The Institute of Medical Science, The University of Tokyo, Tokyo, Japan, 9 International Research Center for Infectious Diseases, The Institute of Medical Science, The University of Tokyo, Tokyo, Japan, 10 Graduate School of Frontier Sciences, The University of Tokyo, Kashiwa, Japan, 11 CREST, Japan Science and Technology Agency, Kawaguchi, Japan, 12 Center for Infectious Disease Education and Research, Osaka University, Suita, Japan, 13 Institute of Mathematics for Industry, Kyushu University, Fukuoka, Japan, 14 Institute for the Advanced Study of Human Biology (ASHBi), Kyoto University, Kyoto, Japan, 15 NEXT-Ganken Program, Japanese Foundation for Cancer Research (JFCR), Tokyo, Japan, 16 Interdisciplinary Theoretical and Mathematical Sciences Program (iTHEMS), RIKEN, Saitama, Japan, 17 Science Groove Inc., Fukuoka, Japan, 18 AMED-CREST, Japan Agency for Medical Research and Development (AMED), Tokyo, Japan

☉ These authors contributed equally to this work.
¤a Current address: Insect-Virus Interactions Unit, Department of Virology, Institute Pasteur, Paris, France
¤b Current address: Department of Scientific computing, Pukyong National University, Busan, South Korea
¶ Membership of The Genotype to Phenotype Japan (G2P-Japan) Consortium is provided in Supporting Information file [S1 Acknowledgement].
* matsuura@biken.osaka-u.ac.jp (YM); shimamura@med.nagoya-u.ac.jp (TS); iwami.iblab@bio.nagoya-u.ac.jp (SI); fukut@pop.med.hokudai.ac.jp (TF)

**Data Availability Statement:** All relevant data are within the manuscript and its Supporting information files.

## Abstract

Mutations continue to accumulate within the SARS-CoV-2 genome, and the ongoing epidemic has shown no signs of ending. It is critical to predict problematic mutations that may arise in clinical environments and assess their properties in advance to quickly implement countermeasures against future variant infections. In this study, we identified mutations resistant to remdesivir, which is widely administered to SARS-CoV-2-infected patients, and discuss the cause of resistance. First, we simultaneously constructed eight recombinant viruses carrying the mutations detected in *in vitro* serial passages of SARS-CoV-2 in the presence of remdesivir. We confirmed that all the mutant viruses didn't gain the virus production efficiency without remdesivir treatment. Time course analyses of cellular virus infections showed significantly higher infectious titers and infection rates in mutant viruses than wild type virus under treatment with remdesivir. Next, we developed a mathematical model in consideration of the changing dynamic of cells infected with mutant viruses with distinct propagation properties and defined that mutations detected in in vitro passages canceled

**Funding:** This study was supported in part by a Grant-in-Aid for JSPS Scientific Research (KAKENHI) (21H02736 to TF, 19K24679 to TF, 18KT0018 to SI, 18H01139 to SI, 16H04845 to SI, 20H04281 to TS); Scientific Research in Innovative Areas (20H05042 to SI, 19H04839 to SI, 18H05103 to SI, 20H04841 to TS); AMED CREST (19gm1310002 to SI, JP22gm1610008 to TF); AMED Japan Program for Infectious Diseases Research and Infrastructure (20wm0225002 to TF, JP20he0822006 to TF, JP20fk0108264 to TF, JP20he0822008 to TF, JP20wm0225003 to TF, JP20fk0108267 to TF, JP19fk0108113 to TF, JP20wm0125010 to TF, 20wm0325007h0001 to SI, 20wm0325004s0201 to SI, 20wm0325012s0301 to SI, 20wm0325015s0301 to SI); AMED Research Program on Emerging and Re-emerging Infectious Diseases (20fk0108401 to TF, 20fk010847 to TF, 21fk0108617 to TF, 20fk0108451 to TF, 19fk0108050h0003 to SI, 19fk0108156h0001 to SI, 20fk0108140s0801 to SI and 20fk0108413s0301 to SI); AMED Program for Basic and Clinical Research on Hepatitis (19fk0210036h0502 to SI); AMED Program on the Innovative Development and the Application of New Drugs for Hepatitis B (19fk0310114h0103 to SI); JST MIRAI to SI; Moonshot R&D (JPMJMS2021 to SI, JPMJMS2025 to SI); Mitsui Life Social Welfare Foundation to SI; Shin-Nihon of Advanced Medical Research to SI; Suzuken Memorial Foundation to SI; Life Science Foundation of Japan to SI; SECOM Science and Technology Foundation to SI; The Japan Prize Foundation to SI; Daiwa Securities Health Foundation to SI. AMED 20fk0108401 and 20fk010847 were the sources of funding for the construction of all mutant SARS-CoV-2. The funders had no role in study design, data collection and analysis, decision to publish, or preparation of the manuscript.

**Competing interests:** The authors have declared that no competing interests exist.

the antiviral activities of remdesivir without raising virus production capacity. Finally, molecular dynamics simulations of the NSP12 protein of SARS-CoV-2 revealed that the molecular vibration around the RNA-binding site was increased by the introduction of mutations on NSP12. Taken together, we identified multiple mutations that affected the flexibility of the RNA binding site and decreased the antiviral activity of remdesivir. Our new insights will contribute to developing further antiviral measures against SARS-CoV-2 infection.

## Author summary

Considering the emerging Omicron strain, quick characterization of SARS-CoV-2 mutations is important. However, owing to the difficulties in genetically modifying SARS-CoV-2, limited groups have produced multiple mutant viruses. Our cutting-edge reverse genetics technique enabled construction of eight reporter-carrying SARS-CoV-2 with the mutations detected in *in vitro* serial passages of SARS-CoV-2 in the presence of remdesivir. We confirmed that all the mutant viruses didn't gain the virus production efficiency without remdesivir treatment. We developed a mathematical model taking into account sequential changes and identified antiviral effects against mutant viruses with differing propagation capacities and lethal effects on cells. In addition to identifying the positions of mutations, we analyzed the structural changes in SARS-CoV-2 NSP12 by computer simulation to understand the mechanism of resistance. This multidisciplinary approach promotes the evaluation of future resistance mutations.

## Introduction

Severe acute respiratory syndrome coronavirus 2 (SARS-CoV-2) was first discovered in 2019 and quickly spread around the world [1]. Novel SARS-CoV-2 variants have since continued to emerge and the number of virus-infected cases repeats increases and decreases [2]. The clinical spectrum of SARS-CoV-2 infection ranges from mild to critical. While most infections present mild or minor symptoms (e.g. fever, cough, sore throat, malaise, headache, muscle pain, nausea, vomiting, diarrhea, loss of taste and smell), severe acute respiratory disease requires admission to intensive care [3–5]. The illness can be observed even after successful vaccination [6]. Antiviral drugs that can be administered to patients after moderate or severe clinical symptoms have been observed have played important roles in clinical environments. Therefore, it is vital to understand the effectiveness of currently approved antivirals from multiple angles to develop future drugs. In particular, the potential to drive drug resistance should be evaluated because drug-resistant mutations have been observed in several viruses such as influenza A virus, human immunodeficiency virus and hepatitis B virus in the clinical environment [7–10].

Remdesivir (RDV) (GS-5734) is the US Food and Drug Administration (FDA)-approved drug for treatment of coronavirus disease 2019 (COVID-19) patients [11,12]. The compound is an intravenously administered adenosine analogue prodrug that binds to the viral RNA-dependent RNA polymerase and inhibits viral replication. It has demonstrated antiviral activities against a broad range of RNA viruses including Ebolavirus, SARS-CoV, MERS-CoV and SARS-CoV-2 [13–17]. RDV has been widely used in the treatment of SARS-CoV-2 patients, however only two amino acid mutations (D484Y and E802D in non-structural protein [NSP] 12) were identified from SARS-CoV-2 patients that were administered RDV [18,19]. One mutation (E802D) was also found in *in vitro* serial passages of the virus under treatment of

RDV [20]. Although studies regarding E802D revealed that the mutation decreased viral susceptibility to RDV [19,20], the mechanisms of how resistance arises have not yet been analyzed in detail. It is critical to elucidate the mechanisms of RDV resistance and to identify further RDV-resistant mutants that may arise in the future to circumvent resistance mutations before they become established in circulating strains.

To evaluate the effect of each gene mutation on viral propagation, genetically modified viruses should be engineered using the reverse genetics system. We recently established a quick reverse genetics system for SARS-CoV-2 using the circular polymerase extension reaction (CPER) method [21]. Nine viral genome fragments, which cover the full-length viral genome, and a linker fragment that encodes the promoter sequence were amplified by PCR and connected to obtain the circular viral DNAs by an additional PCR. By direct transfection of the circular DNAs, infectious SARS-CoV-2 was rescued. Introduction of reporters or mutations can be quickly completed by overlapping PCR or plasmid mutagenesis using the desired gene fragments of less than 5,000 base pairs (bp). While other reverse genetics systems for SARS-CoV-2 require specific techniques such as *in vitro* transcription or *in vitro* ligation, which are obstacles to mutagenesis [22,23], our method does not need these and has already been applied to the characterization of several viral mutations observed in the different SARS-CoV-2 variants [24,25], allowing us to simultaneously generate multiple mutants [26].

In this study, we attempted to identify multiple RDV-resistant mutations and examine the mechanisms of RDV resistance by a multidisciplinary approach that integrates state-of-the-art reverse genetics, mathematical modeling, and molecular dynamics analyses. We first predicted the presumed RDV-resistant mutations by *in vitro* passages of SARS-CoV-2 in the presence of RDV. Next, the recombinant viruses carrying the predicted mutations were generated by the CPER method and the efficiency of infectious virus production and antiviral effects of RDV on the mutants were examined by mathematical modeling. Finally, the conformational changes of NSP12 induced by mutations were analyzed by molecular dynamics simulations to understand the mechanisms of RDV resistance.

## Results

### *In vitro* serial passages of SARS-CoV-2 in the presence of remdesivir

To identify the genes presumably involved in RDV resistance, we first passaged the SARS-CoV-2 strain SARS-CoV-2/Hu/DP/Kng/19-020 (GenBank: LC528232.2) in HEK293-C34 cells under treatment with RDV (Fig 1A). In order to assess many viral mutations responsible for RDV resistance, we treated the cells with RDV at the highest concentration that would allow the virus to be passed on to the next generation. First of all, the cells were treated with 0.1 μM RDV, however cytopathic effect (CPE) was not observed in 14 days. The cells were then treated with 0.01 μM RDV 0.01 μM and CPE was observed at day 4. The culture supernatants were collected, stored as a Passage 1 (P1) sample (Fig 1A). The concentration of RDV was gradually increased from 0.01 μM to 4.0 μM over 10 passages. Throughout the passages, the virus-infected cells were cultured until CPE was observed (3–8 days). Whereas no CPE was seen for 14 days after 0.1 μM RDV treatment in original viruses (P0), CPE was observed throughout the wells in the presence of 4.0 μM RDV at P10, indicating that the virus decreased susceptibility to RDV during the passages.

After 10 passages with RDV, the culture supernatants were collected (P10 with RDV) and subjected to MiSeq sequencing to determine the full-length viral sequence (DDBJ Accession number: LC742929). We prepared the viruses passaged 10 times without RDV as a control (P10 without RDV). Comparison of the P10 with RDV virus sequence with the original SARS-CoV-2 genome found 14 unique mutation sites and 6 mutation sites in P10 without

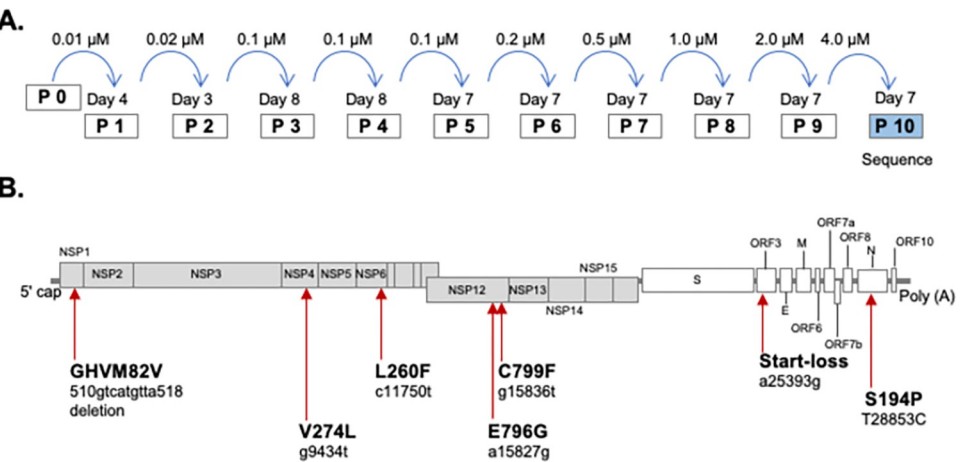

**Fig 1. Identification of RDV-resistant mutations of SARS-CoV-2. (A)** Schematic image of the *in vitro* serial passage of SARS-CoV-2 in the presence of RDV. Supernatants of virus-infected cells were passaged with gradually increased concentrations of RDV. The virus sequence was examined after 10 passages. **(B)** The SARS-CoV-2 genome with the locations of mutations, which demonstrated more than 80% of frequency in Miseq sequence and more than 90% of frequency in single virus sequence after 10-time passages of SARS-CoV-2 with RDV.

RDV virus sequence (Fig 1B and S1 Table). Five mutations found in P10 without RDV had a less than 50% mutant frequency, and mutations were not found in NSP7, 8, 12 nor 13, which are involved in the formation of the replication complex. Importantly, amino acid substitutions E796G and C799F in NSP12 were observed only in the P10 with RDV virus and there have been no reports of these two mutations to date, according to Nextstrain [27]. We then conducted the Sanger sequencing of the viruses after the limiting dilution cloning to investigate whether mutations are introduced in the same virus genomes or not. In total of 10 clones were obtained by the limiting dilution cloning after 10 times passage of SARS-CoV-2 with RDV (P10 with RDV). The deletion of nine nucleotides in NSP1 (82GHVM85V) was observed in 9 single viruses, and amino acid substitutions in NSP4 (V294L), NSP6 (L260F) and NSP12 (E796G and C799F) were observed in all 10 viruses, indicating that these mutations had been introduced in a single virus together (S2 Table). In addition, these mutations demonstrated more than 80% of frequency by Miseq sequence. According to Nextstrain, the same mutations had been detected in NSP1, NSP4 and NSP6, but only a few cases of each mutation had been reported. Besides, the deletion of 15 nucleotides in S, start-loss of ORF3A and S194P in N were observed in every single virus and detected by MiSeq sequence as well and these mutations have not been reported in Nextstrain.

## Generation of RDV-resistant SARS-CoV-2

We then generated high-affinity NanoBiT (HiBiT)-carrying recombinant SARS-CoV-2 with each mutation to identify the RDV-resistant mutations based on strain JPN/TY/WK-521 (GISAID accession number: EPI_ISL_408667). NanoLuc enzymatic activity can be detected by interaction of HiBiT and large NanoBiT (LgBiT), which constitute a split reporter. The reporter SARS-CoV-2 can be generated by inserting only 11 amino acids into the viral genome, and HiBiT-carrying viruses exhibit similar growth kinetics to wildtype (WT) virus [21]. All recombinant SARS-CoV-2 with HiBiT and mutations were prepared using the CPER method that was previously established by our group. Amino acid substitutions were introduced by overlapping PCR and the full-length sequences of the mutant viruses were confirmed prior to assay by Sanger sequencing.

**Table 1. Generation of the recombinant SARS-CoV-2 carrying HiBiT reporter and mutations.**

| HiBiT-viruses | Mutation | Identified from | |
|---|---|---|---|
| WT | No | | |
| E796G | NSP12 E796G | *In vitro* passage of SARS-CoV-2 with remdesivir | This study |
| C799F | NSP12 C799F | | |
| R10/ E796G/ C799F | NSP1 82GHVM85V; NSP4 V274L; NSP6 L260F; NSP12 E796G and C799F | | |
| R10/C799F | NSP1 82GHVM85V; NSP4 V274L; NSP6 L260F; NSP12 C799F | | |
| E802D | NSP12 E802D | SARS-CoV-2-infected patients with the treatment of remdesivir | Szemiel AM et al., Plos Pathogens, 2021; Gandhi S et ai., MedXiv, 2021 |
| D484Y | NSP12 D484Y | | Martinot M et al., Clin Infect Dis., 2021 |
| F480L | NSP12 F480L | *In vitro* passage of MHV with remdesivir | Agostini ML et al., mBio, 2018 |
| V557L | NSP12 V557L (failed to generate) | | |
| F480L/V557L | NSP12 F480L and V557L | | |

Because RDV acts as a nucleoside analog and targets the RNA-dependent RNA polymerase (RdRp) of coronaviruses, including SARS-CoV-2, in the current study we focused on the mutations in NSP12. We generated recombinant viruses with the E796G or C799F mutations that were observed in our P10 serial virus passage (Table 1). We also prepared recombinant viruses with mutations which demonstrated more than 80% of frequency in MiSeq sequence and was common in more than 9 out of 10 single virus clones after P10 (R10/E796G/C799F) or with mutations except for E796G (R10/C799F). In addition to the mutations observed in this study, we also characterized mutations that have been reported as, or anticipated to be, resistant to RDV, as listed in Table 1.

The amino acid mutation E802D in NSP12 was found during the serial passage of SARS-CoV-2 *in vitro* in the presence of RDV and another report showed that the same mutation was found in patients receiving RDV [19,20]. The D484Y mutation was also identified in a COVID-19 patient receiving RDV treatment [18]. Previously, amino acid substitutions F476L and V553L were identified as RDV-resistant mutations in the *Betacoronavirus* murine hepatitis virus (MHV)*(15)*. The two affected amino acid residues (476F and 553L in MHV) are conserved across coronaviruses and correspond to 480F and 557V in the SARS-CoV-2 genome. We attempted to generate recombinant SARS-CoV-2 with either or both mutations but the virus with the single V557L mutation could not be rescued.

We then examined the sensitivity of recombinant viruses to RDV (S1 Fig). Viruses were cultured in the presence of RDV at 0–1.0 μM final concentration for 48 hours and luciferase activity was measured and normalized against control without RDV treatment (0 μM final concentration). The 50% effective concentration ($EC_{50}$) was calculated using the drc package (v3.0–1; R Project for Statistical Computing). All tested mutant viruses showed greater $EC_{50}$ than WT virus, although the difference between WT and F480L mutant was small, indicating that the mutations observed in NSP12 led to decreased viral sensitivity to RDV.

## Time course analyses of infection with presumed RDV-resistant mutants

To investigate the growth efficiency of mutant viruses and the antiviral effects of RDV, we first performed time course analyses of infectious virus production with or without RDV for 96 hours and demonstrated the data up to 72 hours post infection (hpi) because most of the cells have been died and the viral titer have been decreased (Fig 2A and 2B). At all the indicated time points, the infectious titer of each mutant virus was similar or lower than that of the WT

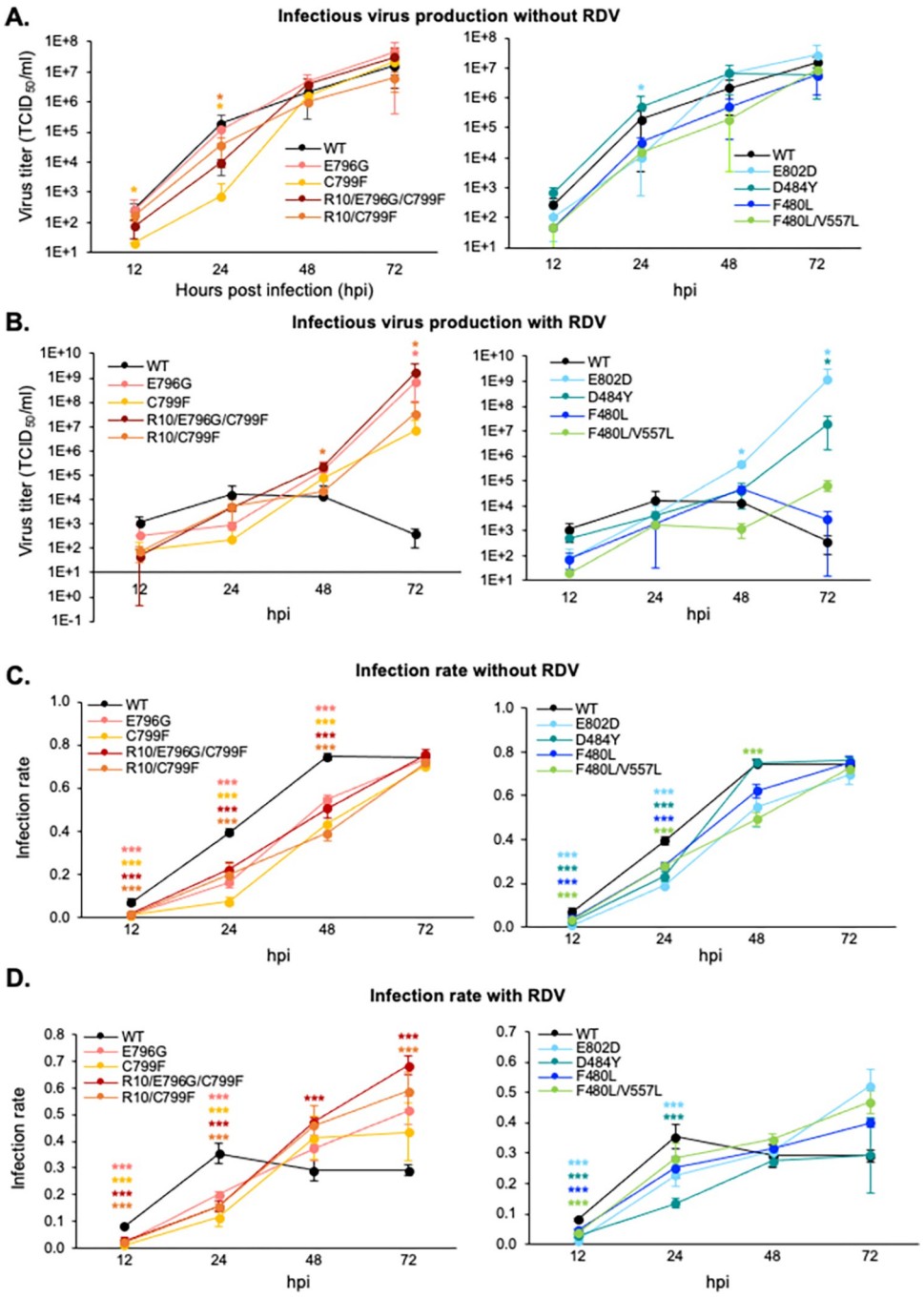

**Fig 2. Infection kinetics of recombinant SARS-CoV-2 with NSP12 mutations. (A and B)** Infectious virus production in the absence **(A)** and presence **(B)** of RDV. Supernatants of virus-infected cells were collected from 12–96 hpi and virus titers were determined by titration. Statistical significances were determined by Kruskal–Wallis test with the two stage linear step-up procedure of Benjamini, Kreiger and Yekutieli. Significant differences compared with WT virus are indicated by an asterisk (*$p < 0.05$). **(C and D)** Infection rate in the absence **(C)** and presence **(D)** of RDV. Virus-infected cells were fixed and stained with antibodies for SARS-CoV-2 and cell nuclei. Infection rates were calculated by dividing the numbers of virus-positive cells by the numbers of nuclei. Statistical significances were determined by one-way ANOVA with Dunnett's test. Significant differences compared with WT virus are indicated by asterisks (***$p < 0.001$).

virus in the absence of RDV treatment, indicating that the mutant viruses produced the infectious viruses with the same or lower efficiency as WT virus (Fig 2A). Conversely, significant differences were observed in the infectious titers of mutant viruses in the presence of RDV at 0.05 μM final concentration. In the left panel of Fig 2B, the infectious titers of mutant viruses (E796G, C799F, R10/E796G/C799F, and R10/C799F) gradually increased for 48 hours and were significantly higher than that of WT virus at 72 hpi. In the right panel of Fig 2B, the infectious titer of the E802D mutant virus increased rapidly and was significantly higher than that of WT virus at 48 and 72 hpi. The titers of the F484Y and F480L/V557L mutants were also significantly higher than that of WT virus. Meanwhile there were no differences between the titers of F480L mutant and WT viruses at the indicated time points. These results suggest that the sensitivity of all the mutant viruses, except for the F480L virus, to RDV was diminished, which was consistent with the results of the RDV susceptibility test demonstrating minimal change in the $EC_{50}$ of the F480L virus (S1 Fig).

Next, we investigated the the ratio of the virus-infected cells (Fig 2C and 2D). HEK293-C34 cells were infected with mutant viruses with and without RDV treatment. Virus-infected cells were harvested and fixed from 12–96 hpi and subjected to immunofluorescent assay using anti-SARS-CoV-2 NP antibody and DAPI. The virus infection rates were then calculated and demonstrated up to 72 hpi. All the mutant viruses demonstrated equivalent or significantly lower virus infection rates compared with WT virus in the absence of RDV. These data suggested that the number of cells infected with mutant viruses increased more slowly compared with WT virus in the absence of RDV treatment, consistent with the data on production of infectious virus particles. Meanwhile, the infection rates of the presumed RDV-resistant mutant viruses, except for D484Y, were higher or significantly higher (R10/E796G/C799F at 48 and 72 hpi, and R10/C799F at 72 hpi) than those of WT virus at 48 and 72 hpi in the presence of RDV, indicating that these mutant viruses can spread more efficiently than WT virus in the presence of RDV.

## Antiviral effect of RDV on the presumed RDV-resistant mutants, analyzed by mathematical modeling

To quantify the kinetic parameters of SARS-CoV-2 and the antiviral effect of RDV on WT and RDV-resistant viruses, we developed a mathematical model for SARS-CoV-2 infection under RDV treatment. We examined the growth rate of HEK293-C34 cells up to 48 hours after seeding (S2A Fig), the degradation rate of SARS-CoV-2 at 37˚C (S2B Fig), the infectious virus production rate for 96 hours (Fig 2A and 2B), and the rate of infection in susceptible cells (Fig 2C and 2D). These estimated parameters were fixed and used here.

To consider the variability of kinetic parameters and model predictions, we performed Bayesian estimation for the whole dataset using Markov chain Monte Carlo (MCMC) sampling, and simultaneously fit Eqs (1–4) with RDV ($\boldsymbol{\varepsilon} > \boldsymbol{0}$) and without RDV ($\boldsymbol{\varepsilon} = \boldsymbol{0}$) to the concentrations of target cells, infected cells, and infectious virus (see Materials and Methods and S3 Fig). The estimated parameters are listed in Tables 2 and 3, and the simulation results of the model using these best-fit parameter estimates are shown with the data in S3 Fig. Comparing the virus production rate and the antiviral effect for the eight different resistant mutants (D484Y, F480L, F480L/V557L, E796G, C799F, R10/C799F, R10/E796G/C799F, E802D), we found that the virus production for all mutants was lower than that of the WT virus except for D484Y mutant (Fig 3B and Table 3). The D484Y mutation had a competitive advantage in virus production rate, and this property might be involved in its decreased susceptibility to RDV, although the difference from WT was only 1–1.25-fold.

**Table 2. Estimated kinetic parameters and initial values.**

| Parameters or variables | Symbol | Unit | Mean | CI 95% |
|---|---|---|---|---|
| **Estimated parameters from separate experiments** | | | | |
| Proliferation rate of target cell and eclipse phase cell | $g$ | day$^{-1}$ | 1.695 | — |
| Carrying capacity of total cell | $K$ | Cells/ml | $2.38\times10^5$ | — |
| Clearance rate of infectious virus | $c$ | day$^{-1}$ | 8.44 | — |
| **Parameters obtained from simultaneous fitting of full in vitro data set** | | | | |
| Rate for infections | $\beta$ | (TCID50/ml•Day)$^{-1}$ | $1.39\times10^{-6}$ | $(0.95–1.97)\times10^{-6}$ |
| Death rate of virus-producing cells | $\delta$ | day$^{-1}$ | 0.42 | (0.32–0.53) |
| Length of eclipse phase | $1/k$ | day | 0.55 | (0.50–0.61) |
| Production rate of infectious virus | $p$ | day$^{-1}$ | $7.38\times10^2$ | $(5.76–9.28)\times10^2$ |
| Initial value for target cells | $T(0)$ | Cells/ml | $3.95\times10^5$ | $(2.34–6.74)\times10^5$ |
| Initial value for virus-producing cells | $I(0)$ | Cells/ml | 0.23 | (0.04–0.78) |
| Initial value for infectious virus | $V(0)$ | TCID50/ml | $3.99\times10^4$ | $(1.86–7.33)\times10^4$ |

Interestingly, all the other tested mutations strongly suppressed virus production. RDV showed more than 70% antiviral effect on three mutations, 81% (95% CI: 69–89) for D484Y; 80% (95% CI: 70–87) for F480L; and 78% (95% CI: 71–86) for F480L/V557L. However, the antiviral effect on two mutations was less than 40%, at 38% (95% CI: 29–49) for E802D and 38% (95% CI: 27–54) for R10/E796G/C799F (Fig 3C and Table 3). All the examined mutations reduced the antiviral effect of RDV and the change was greatest in the mutations found in *in vitro* passages. The antiviral effect of RDV in the F480L mutation was not much different from that observed with WT virus, which is consistent with the results of the RDV susceptibility test. Because there was no correlation between the efficiency of virus production and the antiviral effect, the mechanisms of RDV resistance were predicted to be a structural defect in the direct interaction between the viral genome replication complex and RDV.

## RDV-resistance mechanisms of the presumed RDV-resistant mutants, analyzed by computer simulation

Molecular modeling and molecular dynamics simulations were performed to clarify the structural and property changes caused by amino acid mutations in the NSP12 protein. The representative complete structure of the prepared protein-RNA complex is shown in Fig 4A. In this figure, NSP12, binding RNA, and RDV are shown in cartoon, ball and stick, and van der Waals notation, respectively. RDV is located at the end of the binding RNA and is inside the protein. Then, the root mean square deviations (RMSDs) were compared using molecular dynamics simulation trajectories of each complex of WT or mutant NSP12 protein and RNA-incorporated RDV, as shown in Fig 4B. Most complexes reached thermodynamic stability and

**Table 3. Estimated parameters of the virus production and the antiviral effect for eight different resistant mutants and wild type.**

| Parameters or variables (symbol) | Wild type | E796G | C799F | R10/E796G/ C799F | R10/C799F | E802D | D484Y | F480L | F480L/ V557L |
|---|---|---|---|---|---|---|---|---|---|
| Fold-change of the production rate of infectious virus ($\eta$) | 1 (—) | 0.82(0.64 −1.02) | 0.29(0.21 −0.40) | 0.54(0.41 −0.69) | 0.51(0.37 −0.71) | 0.55(0.40 −0.73) | 1.10(0.94 −1.26) | 0.48(0.39 −0.58) | 0.44(0.31 −0.60) |
| Inhibition rate of virus production by remdesivir ($\varepsilon$) | 0.92[†](0.87 −0.95)[‡] | 0.57(0.25 −0.77) | 0.50(0.34 −0.66) | 0.38(0.27 −0.54) | 0.58(0.41 −0.75) | 0.38(0.29 −0.49) | 0.81(0.69 −0.89) | 0.80(0.70 −0.87) | 0.78(0.71 −0.86) |

[†] Mean value

[‡] 95% confidence interv

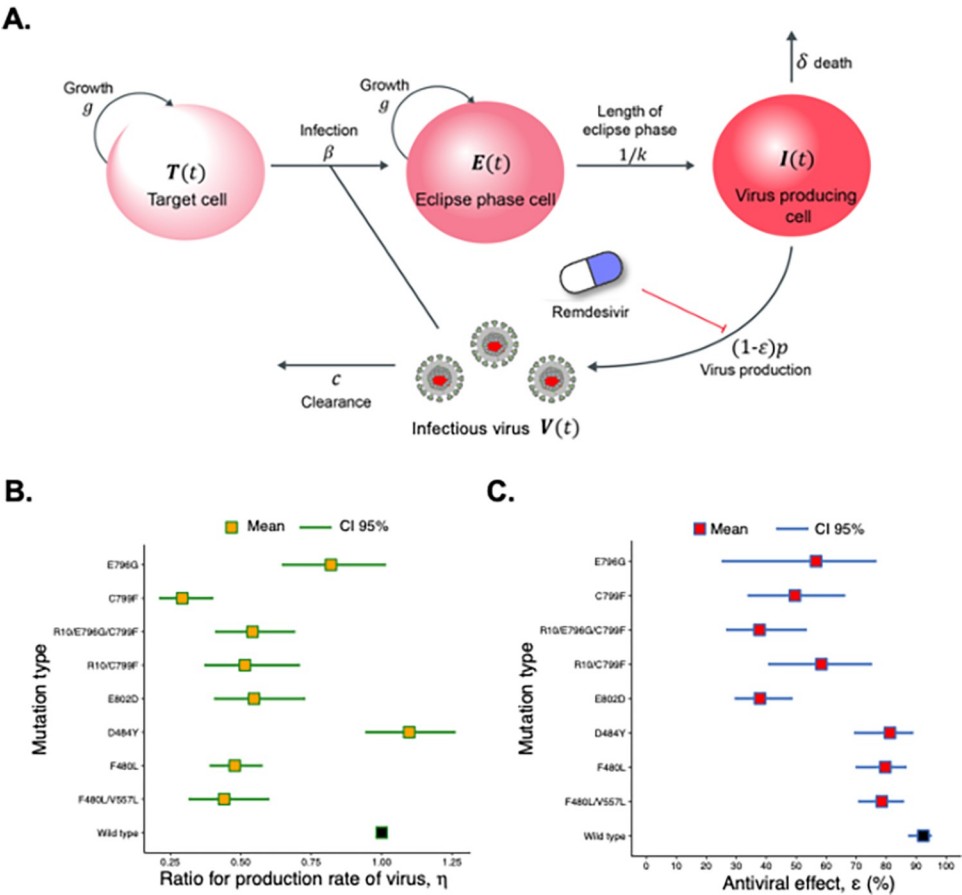

**Fig 3. Quantifying antiviral effects of RDV against SARS-CoV-2 infection. (A)** Schematic diagram of SARS-CoV-2 infection under RDV treatment in cell culture. The target cells are infected by infectious virus at rate **β** and become eclipse phase cells. The average duration of the eclipse phase is **1/k** days and these eclipse phase cells start producing viruses at rate **p** (i.e., become virus-producing cells). The target cells and eclipse phase cells are assumed to divide at rate **g**, and virus-producing cells die at rate **δ**. Progeny infectious viruses are cleared at rate **c**. RDV blocks virus production by inhibiting viral replication in infected cells with inhibition rate **ε**. **(B)** Comparison of the fold-change of virus production rate, **η**, for eight different RDV-resistant viruses and WT virus. Orange dots are the mean value and green lines show 95% CI, which were estimated from the experiments with/without RDV. For WT, we consider **η = 1**, shown by the black dot. **(C)** Comparison of the antiviral effect of RDV, **ε**, for eight different RDV-resistant viruses and WT virus. Red and black dots are the mean values of the antiviral effects and blue lines show 95% CI, which were estimated from infection experiments with/without RDV.

plateau in RMSD after 200,000 steps. However, only the V557L mutant structure failed to reach the stabilized structure. When RNA structures for this mutant were superimposed and RMSDs were calculated for RNA and protein separately, the increase in RMSDs for RNA reached a plateau, while the RMSDs for protein continued to increase as before. This behavior means the bonds between protein and RNA tended to move apart. In other words, the RNA-protein complex tended to be unstable, which may correspond with the inability of this mutant virus to multiply, indicating the probable reason for failed rescue of the V557L mutant recombinant SARS-CoV-2 (Table 1). Therefore, we proceeded with the analysis of the other mutants only.

Using the trajectories obtained from molecular dynamics calculations, we calculated and compared the variation of RMSD for each substructure. The RMSDs of each mutant complex and the WT complex were searched for areas where they differed significantly, and the results are shown in Fig 4C. Locations where the RMSD variation of the mutant is greater than that of

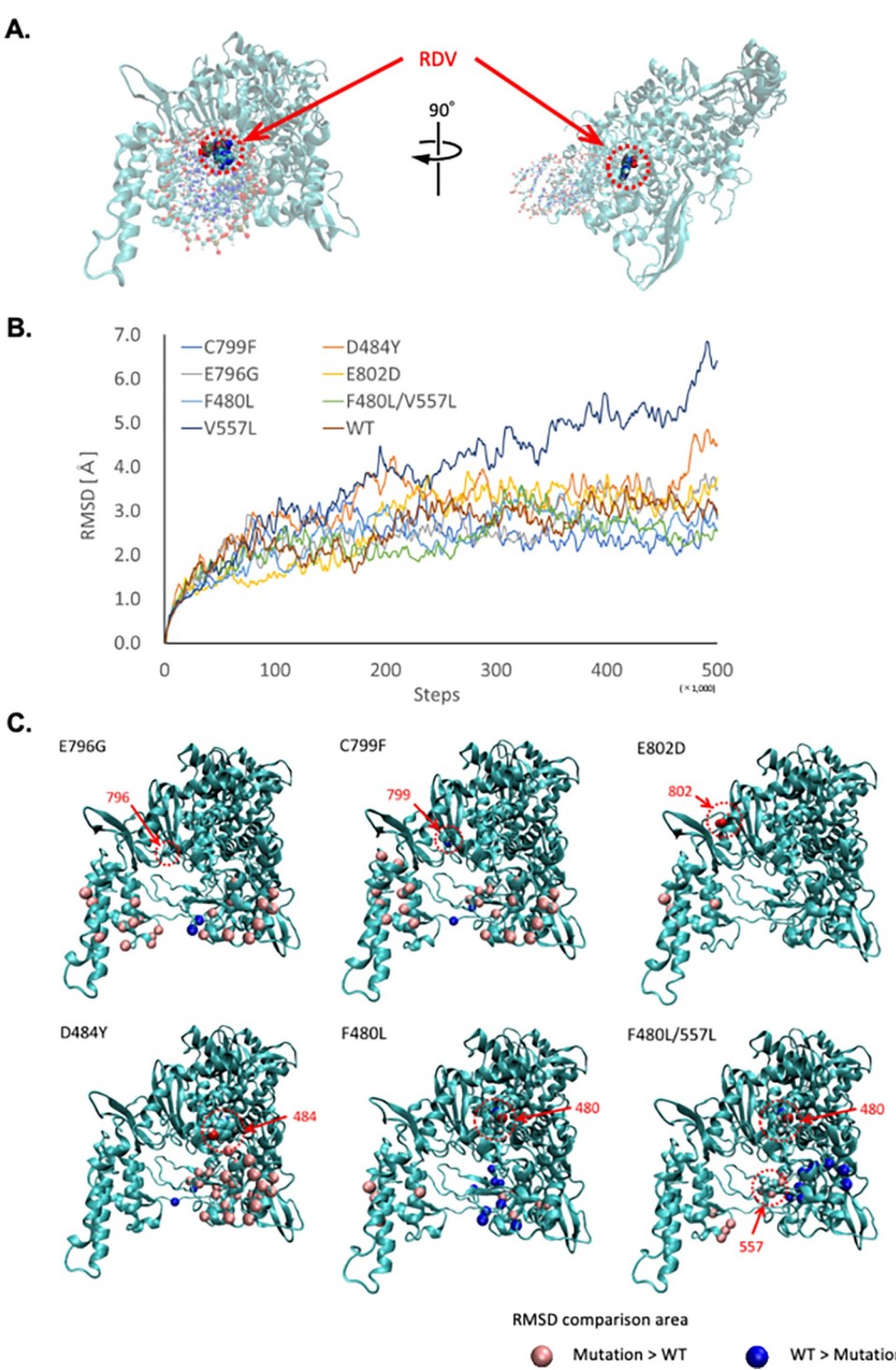

**Fig 4. NSP12-RNA binding structure and comparison of thermodynamic stability. (A)** Overall view of NSP12 protein and location of the RDV. **(B)** RMSD comparison of RNA-binding proteins. **(C)** Comparison of the molecular vibrations of WT and each mutant. The pink and blue spheres represent regions of large oscillations in the mutant and WT, respectively.

the WT are indicated by pink spheres, and conversely, locations where the RMSD variation of the WT is greater than that of the mutant are indicated by blue spheres. In the tested mutations, except for F480L, the molecular vibration of the mutants tended to increase around the RNA-binding site as shown in S4 Fig, indicating that introduction of the mutations increased the flexibility of the RNA binding site. However, in the center of the RNA-binding site (near the RDV-binding site) of the F480L mutant, the molecular vibration of the mutant tended to be small, which is consistent with the small change in antiviral effect (Fig 3C) and RDV sensitivity (S1 Fig).

Taken together, NSP12 mutations found in previous studies and in our *in vitro* virus passages decreased the antiviral effect of RDV, though not to the same degree, and influenced increased flexibility of the RNA-binding site of the NSP12 structure.

## Propagation efficiency of the presumed RDV-resistant mutants in *in vivo* Syrian hamster

Finally, we also examined the pathogenicity and viral growth kinetics of WT virus and virus with RDV-resistant mutations in *in vivo* hamster (Fig 5). The hamster group inoculated with WT virus showed significant weight loss compared to hamster inoculated with R10/E796G/C799F virus at 4–7 dpi (Fig 5B). The oral swab samples were collected, and the viral RNA copies were compared for 5 days. In each time point, the viral RNA copies were significantly lower in the hamster group inoculated with R10/E796G/C799F virus. These findings indicate that R10/E796G/C799F virus has a less viral propagation efficiency in hamster than WT virus, which is consistent with the lower virus production rate of R10/E796G/C799F virus in HEK293-C34 cells (Figs 2A, 2C and 3B).

## Discussion

As demonstrated by the recent emergence of the Omicron strain, mutations have continuously accumulated on the SARS-CoV-2 genome [28]. To implement the most effective measures against COVID-19, it is crucial to predict clinically important mutations in advance, including drug resistance mutations. In this study, we generated eight recombinant viruses with presumed RDV-resistant mutations, using a simple reverse genetics system. Quantifying the kinetic parameters for RDV-resistant viruses showed no dramatic increases in the efficiency of infectious virus production in any mutant virus. Besides, R10/E796G/C799F virus was less efficient in virus propagation in hamsters than WT virus. But importantly, all mutants rendered RDV ineffective to some extent. Molecular modeling and molecular dynamics simulations of the mutant NSP12 proteins revealed that the tested mutations, excluding F480L, contributed to increased molecular vibrations around the RNA-binding site. Our multidisciplinary approach of molecular virology, mathematical, and molecular modeling discovered mutations involved in RDV resistance and the mechanisms of this drug resistance.

When evaluating virus growth kinetics, titers of different viruses are generally compared at the same time points. However, the total numbers of cells susceptible to viral infection are not constant between wells. The faster a virus propagates and spreads, the faster the number of cells available for virus infection decreases. Thus, comparison of titers at specific time points may not be the best method of evaluating the proliferative capability of viruses with different properties or the effects of drugs against them. Here we combined mathematical models and statistical methods, analyzed intercellular infection dynamics of SARS-CoV-2 in the momently changing cells, and identified the differences in infectious virus production and antiviral effects between WT and mutant viruses (Fig 3).

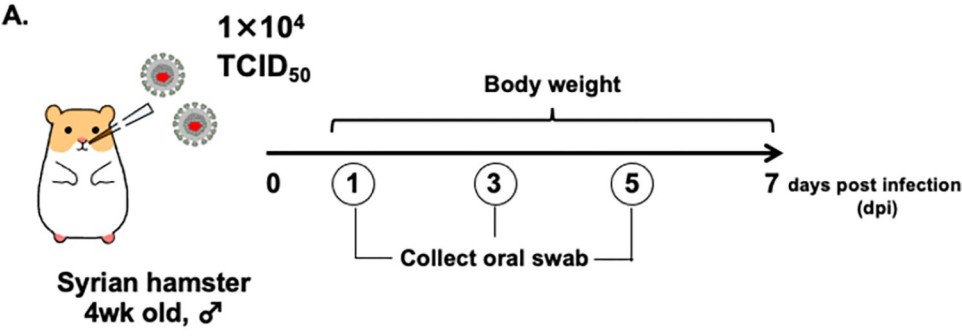

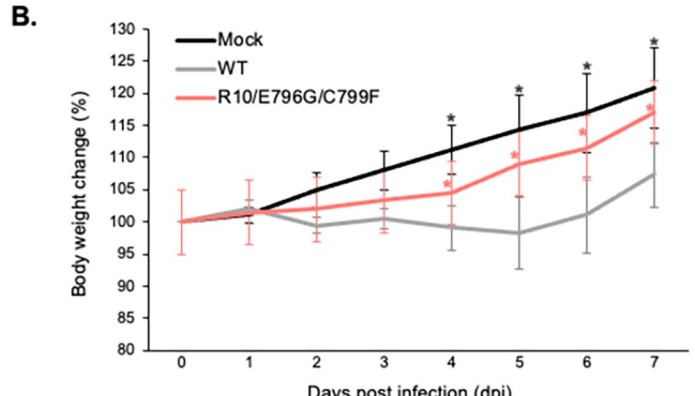

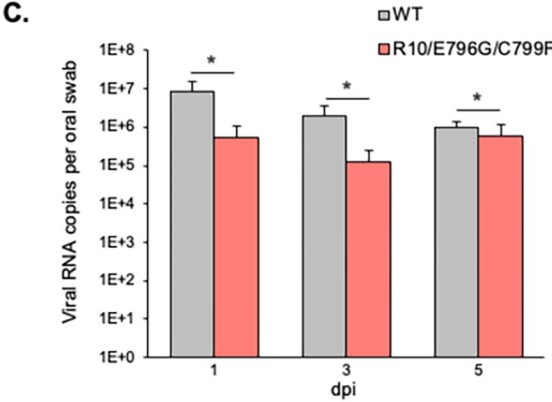

**Fig 5. The pathogenicity and growth kinetics of RDV-resistant SARS-CoV-2 in Syrian hamsters.** (A) Schematic diagram of WT and R10/E796G/C799F viruses infection and animal operations. Syrian hamsters were inoculated with saline (n = 4, uninfected control), WT (n = 6) or R10/E796G/C799F virus (n = 6). Body weight was daily examined and oral swabs were collected at indicated timepoints. (B) Change of body weight. Data are mean ± SD. Statistical significances were determined by Kruskal–Wallis test with the two stage linear step-up procedure of Benjamini, Kreiger and Yekutieli. Significant differences compared with WT virus are indicated by an asterisk (*$p < 0.05$). (C) Viral RNA load in oral swab was measured. Data are mean ± SD. Statistical significances (*$p < 0.05$). were determined by two-tailed Student's t test.

Furthermore, molecular simulation of the NSP12 structure clarified the mechanism causing failure of V557L to proliferate and provided new insights into the mechanisms of RDV resistance of the tested mutants (Fig 4), in contrast to other previous studies that only identified the

location of viral mutations. The SARS-CoV-2 replication-transcription complex indicated in the PDB ID:501 6XEZ (in S5A Fig) and the SARS-CoV-2 NSP12 (in S5B Fig) demonstrated the similar RMSDs (in S5C Fig). Besides, RMSD for the average structure of the whole model after 300,000 steps, which is considered to be equilibrated, was 1.64 Å. These findings indicates that thermodynamic oscillations of NSP12 is not changed largely, and the small model which was used in this study is accurate enough to evaluate the effects of the mutations on molecular vibration. In mutants resistant to RDV, regardless of the location of the mutation, large differences were observed in the RNA binding site when thermodynamic oscillations were compared with WT. Mutations not involved in resistance showed the opposite trend of variation in terms of protein flexibility. We therefore speculate that flexibility in the binding site may be a factor in resistance to RDV. More detailed mechanisms of resistance might be revealed by evaluating polymerase activities of mutants.

Three mutations (C799F, E796G, and E802D) markedly diminished the antiviral effect of RDV (Fig 3C). These mutations did not contribute to efficient virus production but increased the flexibility of the RNA-binding site, which probably enabled these viruses to evade the functional inhibition by RDV binding. The C799F virus was less efficient in virus production than R10/C799F virus (Fig 3B). Besides, the mutations introduced in R10/C799F virus was not detected in the viruses passaged 10 times in absence of RDV by MiSeq sequeincig and observed in the same individual virus genomes (S1 and S2 Tables). Therefore, some mutations, which were introduced in R10/C799F or R10/ E796G/C799F and observed in genome regions other than NSP12, might restore the efficiency of virus production in the mutant viruses. Unlike the mutations found in *in vitro* analyses, D484Y, F480L, and F480L/V557L affected neither virus production nor RDV sensitivity. We expect that even the homologous amino acid mutations (F480L and V537L) would have different effects on the NSP12 of MHV and SARS-CoV-2, since there are also many different amino acid residues within NSP12 that interact with them to change the structure of NSP12. Importantly, these results consistent between *in vitro* passages and computer simulations highlight the accuracy and usefulness of current interdisciplinary research.

In the previous study, 2 amino acid substitutions were observed in NSP12 by *in vitro* passages of MHV with remdesivir and the mutations decreased viral fitness of MHV *in vitro*. Mutant SARS-CoVs with the corresponding mutations also attenuated the pathogenesis in mice [15]. Therefore, we anticipated that the NSP12 mutations observed in *in vitro* passage of SARS-CoV-2 with the presence of remdesivir in this study had neither gained pathogenicity nor fitness and then constructed them by CPER method. Every mutation observed in *in vitro* passages of SARS-CoV-2 failed to increase the efficiency of infectious virus production. Morevover, R10/E796G/C799F virus was less effective than WT virus in propagation in hamsters. Co-infection competition assay in previous studies revealed that E802D on SARS-CoV-2 or F476L/V553L mutations on MHV decreased fitness [15,20], consistent with our results. Although further characterization is necessary, these results indicate that the observed mutations in this study are unlikely to be persistent in the virus population without RDV or dramatically accelerate the SARS-CoV-2 epidemic. However, because the E802D mutation found initially during *in vitro* passage was actually detected in clinical patients [19,20], it is quite possible that the analyzed mutations may be reported following the sustained administration of RDV, and thus continuous viral sequence analyses from SARS-CoV-2 patients treated with RDV is vital. Currently, we cannot definitively state that the repeated administration of RDV will be problematic, but additional compounds with higher affinity for the RdRp complex than RDV are likely to become desirable.

The SARS-CoV-2 pandemic is not coming to an end, rather, new virus strains have sequentially emerged. Deletions and mutations that can facilitate higher transmissibility or antibody

evasion have been highlighted in the Omicron variant [29–33]. It has been revealed that two doses of mRNA vaccine are insufficient to provide immunity against the Omicron variant [34–36], and new vaccines that will provide more robust immunity are urgently needed. In addition, new antivirals have been developed and are expected to be approved for clinical use [37–39]. While development of new vaccines and antivirals will continue, it remains important to evaluate their safety, and to pay attention to resistant mutations. Our state-of-the-art study, in which the drug sensitivities of multiple mutant viruses were simultaneously determined, will accelerate the development of new measures, evaluation of drug resistance and deepen our understanding of the driving forces for mutation of SARS-CoV-2.

## Materials and methods

### Cells and viruses

HEK293-C34 cells were previously established and a different clone than HEK293-3P6C33 cell, both of which were IFNAR1 deficient, with expression of human ACE2 and TMPRSS2 induced by doxycycline hydrochloride [21]. The HEK293-C34 cells were maintained in Dulbecco's Modified Eagle Medium (DMEM) (Nacalai Tesque) containing 10% fetal bovine serum (FBS) (Sigma) and blasticidin (10 μg/ml) (Invivogen). The exogenous expression of ACE2 and TMPRSS2 in the HEK293-C34 cells was induced by addition of doxycycline hydrochloride (1 μg/ml) (Sigma). TMPRSS2-expressing Vero E6 (VeroE6/TMPRSS2) cells were purchased from the Japanese Collection of Research Bioresources Cell Bank (JCRB1819) and maintained in DMEM containing 10% FBS and G418 (Nacalai Tesque). Both HEK293-C34 cells and VeroE6/TMPRSS2 cells were cultured at 37˚C in 5% $CO_2$.

SARS-CoV-2 strain SARS-CoV-2/Hu/DP/Kng/19-020, was kindly provided by Dr. Sakuragi at the Kanagawa Prefectural Institute of Public Health and strain JPN/TY/WK-521 by Dr. Masayuki Shimojima at the National Institute of Infectious Diseases. All experiments involving SARS-CoV-2 were performed in biosafety level 3 laboratories, following the standard biosafety protocols approved by the Research Institute for Microbial Diseases at Osaka University.

### Chemical inhibitors and antibodies

RDV (GS-5734) was purchased from Cayman Chemical, dissolved in dimethyl sulfoxide (DMSO) and stored at 50 mM at −30˚C. To detect SARS-CoV-2-infected cells, mouse monoclonal antibody against SARS-CoV-2 NP (Clone# S2N4-1242) was kindly provided by Bio Matrix research. Alexa Fluor 488-conjugated anti-mouse antibodies were purchased from Life Technologies.

### Serial passages of SARS-CoV-2

SARS-CoV-2 strain SARS-CoV-2/Hu/DP/Kng/19-020 was serially passaged in HEK293-C34 cells 10 times in the presence of RDV (Fig 1). HEK293-C34 cells were prepared with DMEM containing 10% FBS, blasticidin, and 1 μg/ml doxycycline hydrochloride in six-well plates. One day later, the cells were infected with SARS-CoV-2 at a multiplicity of infection (MOI) of 0.01 for 1 hour to allow virus attachment. Culture supernatants were then replaced with DMEM containing 2% FBS, blasticidin, 1 μg/ml doxycycline hydrochloride, and RDV. The virus-infected cells were incubated and the supernatants were collected when CPE was observed throughout the wells. The collected supernatants were centrifuged at 1,500 ×g for 5 min to remove cells and debris, and 10 μl of the supernatants were passaged. The titer of the virus in each passage was demonstrated in S4 Table. The final concentration of RDV was gradually increased from 0.01 μM in P1 to 4.0 μM in P10. As a control, SARS-CoV-2 strain SARS-CoV-2/Hu/DP/Kng/19-020 was also passaged in HEK293-C34 cells 10 times in the absence of

RDV. The virus was infected at a MOI of 0.01 and the culture supernatants were collected at day 2. The collected supernatants were centrifuged at 1,500 ×g for 5 min to remove cells and debris, and 10 μl of the supernatants were passaged.

## Validation of the virus genome sequence using Sanger sequence

Total RNA was extracted from the supernatants of SARS-CoV-2-infected cells by using a Pure-Link RNA mini kit (Invitrogen) and subjected to cDNA synthesis using a PrimeScript RT reagent kit (Perfect Real Time) (TaKaRa Bio) and random hexamer primers. A total of nine DNA fragments, covering the full-length SARS-CoV-2 genome were amplified by PCR using a PrimeSTAR GXL DNA polymerase (TaKaRa Bio), the synthesized cDNA and specific primer sets from CoV-2-G1-Fw to CoV-2-G10-Rv designed previously [21]. The amplified PCR fragments were purified using a gel/PCR DNA isolation system (Viogene) and sequenced in both directions using the ABI PRISM 3130 genetic analyzer (Applied Biosystems) with specific primers for SARS-CoV-2.

## Viral genome sequencing using MiSeq

Viral genome sequencing was performed as previously described [24,25,31,40,41]. Briefly, the virus sequences were verified by viral RNA-sequencing analysis. Viral RNA was extracted using a QIAamp viral RNA mini kit (Qiagen). The sequencing library employed for total RNA sequencing was prepared using the NEBNext Ultra RNA Library Prep Kit for Illumina (New England Biolabs). Paired-end 76-bp sequencing was performed using a MiSeq system (Illumina) with MiSeq reagent kit v3 (Illumina). Sequencing reads were trimmed using fastp v0.21.0 [42] and subsequently mapped to the viral genome sequences of a lineage B isolate (strain SARS-CoV-2/Hu/DP/Kng/19-020) using BWA-MEM v0.7.17 [43]. Variant calling, filtering, and annotation were performed using SAMtools v1.9 [44] and snpEff v5.0e [45].

## Limiting dilution cloning

After the serial passage of SARS-CoV-2 10 times with RDV (P10 with RDV), the virus was prepared at $10^3$ $TCID_{50}$ /ml. Then virus was conducted three-fold serial dilution with DMEM containing 2% FBS and inoculated into VeroE6/TMPRSS2 cells prepared in 96-well plates. After 72 hours, the supernatants were collected from a well where a single focus was observed. The collected supernatants were centrifuged at 3,000 ×g for 5 min to remove cells and debris and the virus genome was validated by the Sanger sequence.

## Rescue of the presumed RDV-resistant viruses

All the HiBiT-carrying SARS-CoV-2 with RDV-resistant mutations were rescued by the CPER method, which was established in our previous study [21]. Briefly, nine cDNA fragments, covering the entire genome of SARS-CoV-2 were prepared by PCR using PrimeSTAR GXL DNA polymerase and SARS-CoV-2 viral gene fragment-encoding plasmids. In addition, an untranslated region (UTR) linker fragment encoding the 3′ 43 nucleotides (nt) of SARS-CoV-2, hepatitis delta virus ribozyme (HDVr), bovine growth hormone (BGH) poly(A) signal, cytomegalovirus (CMV) promoter, and the 5′ 25 nt of SARS-CoV-2, was amplified by PCR. A HiBiT luciferase gene (VSGWRLFKKIS) and a linker sequence (GSSG) were introduced into the N terminus of the ORF6 gene of SARS-CoV-2 by site-directed mutagenesis of the viral genome fragment-cloning plasmid and the plasmid was used as a template to amplify a cDNA fragment. Presumed RDV-resistant mutations were introduced into the SARS-CoV-2 cDNA fragments by overlap PCR using specific overlapping primer sets (S3 Table). The nine

SARS-CoV-2 cDNA fragments and the UTR linker fragment (0.1 pmol each) were mixed together and subjected to CPER. The CPER products were then directly transfected into HEK293-C34 cells using Trans IT LT-1 (Mirus). At 6 hours post-transfection, the culture media were changed to DMEM containing 2% FBS, blasticidin, and doxycycline hydrochloride (1 mg/ml). When CPE was observed throughout the wells (usually around 7 days post-transfection), the culture supernatants were collected and centrifuged at 1,500 ×g for 5 min to remove cells and debris. The culture supernatants were then passaged once using VeroE6/TMPRSS2. The virus sequences were confirmed in the passaged virus solutions by Sanger sequencing and thereafter the virus stocks were stored at −80°C until use. The experimental procedure for the construction of mutant SARS-CoV-2 was approved by the committee on genetically modified organisms in Osaka university (project number 4680) and Japanese Minister of Education, Culture, Sports, Science and Technology in compliance with the Cartagena Protocol on Biosafety (2受文科振第764号). All experiments using mutant SARS-CoV-2 were performed in fully licensed BSL-3 facilities in Osaka university.

### Virus titration

Infectious titers in culture supernatants were determined by 50% tissue culture infective doses (TCID$_{50}$). The TCID$_{50}$ was calculated with Reed–Muench Method [46]. VeroE6/TMPRSS2 cells were prepared in 96-well plates and infected with SARS-CoV-2 after ten-fold serial dilution with DMEM containing 2% FBS. Virus titers were determined at 72 hpi.

### RDV susceptibility analysis using the HiBiT system

HEK293-C34 cells were seeded in 48-well plates in DMEM with 10% FBS, blasticidin and 1 μg/ml doxycycline hydrochloride. One day later, HiBiT-carrying SARS-CoV-2 was allowed to attach for 1 hour. Culture media were then replaced with new media containing 2% FBS, blasticidin, 1 μg/ml doxycycline hydrochloride, and RDV (0–1.0-μM final concentration). At 48 hpi, luciferase activity was measured using a Nano-Glo HiBiT lytic assay system (Promega), following the manufacturer's protocols. Briefly, Nano-Glo substrate including LgBiT protein was added to the virus-infected cell lysates after all culture supernatants were removed. Luciferase activities were measured using a luminometer and normalized to luminescence without RDV treatment (0-μM final concentration). The EC$_{50}$ was calculated using the drc package (v3.0–1; R Project for Statistical Computing).

### Time course analyses of infectious virus production

HEK293-C34 cells were prepared in 96-well plates in media containing 1 μg/ml doxycycline hydrochloride. Cells were infected with HiBiT-carrying viruses at MOI = 0.01 for 1 hour. Culture media were changed to fresh media containing 2% FBS, blasticidin, and 1 μg/ml doxycycline hydrochloride, with or without 0.05 μM RDV (final concentration). At 12, 24, 48, 72, and 96 hpi, culture supernatants of the virus-infected cells were collected and infectious titers in the supernatants (TCID$_{50}$/ml) were determined by virus titration.

### Time course analyses of infection rates in cells

After removal of the culture supernatants at the indicated time points in the time course analyses of infectious virus production, the cells were fixed with 4% paraformaldehyde (Nacalai Tesque). The fixed cells were permeabilized with 0.2% Triton X-100 (Nacalai Tesque) in PBS for 20 min, blocked with 1% bovine serum albumin fraction V (Sigma) in PBS, and then reacted with anti-SARS-CoV-2 NP antibody in PBS for 1 hour at room temperature. After washing

with PBS three times, the cells were incubated with a 1:1,000 dilution of goat anti-mouse IgG Alexa Fluor 488-conjugated secondary antibody (Thermo Fisher Scientific) in PBS for 1 hour at room temperature. The cells were then incubated with DAPI (Thermo Fisher Scientific) (1:2,000 dilution) for 10 min. Immunopositive signals were confirmed under a FluoView FV1000 confocal laser scanning microscope (Olympus), with appropriate barrier and excitation filters. Quantitative imaging data were obtained using a CellVoyager CQ1 benchtop high-content analysis system (Yokogawa Electric Corporation) and analyzed with CellPathfinder high content analysis software (Yokogawa Electric Corporation). The number of SARS-CoV-2-infected cells stained by anti-SARS-CoV-2 NP antibody and the number of cell nuclei stained by DAPI were counted. The infection rates were then calculated by dividing the number of SARS-CoV-2-positive cells by the total number of cell nuclei.

## Growth of HEK293-C34 cells

HEK293-C34 cells were seeded in 48-well plates with DMEM containing 10% FBS and blasticidin. At 24 hours post-seeding, media were replaced with DMEM containing 10% FBS, blasticidin, and 1 μg/ml doxycycline. All the cells were collected, and the cell numbers were counted every 12 hours for 48 hours post-medium change.

## Degradation rate of HiBiT-carrying SARS-CoV-2

HiBiT-carrying SARS-CoV-2 were incubated at 37˚C with 5% $CO_2$ for 48 hours. Every 12 hours, the virus solutions were collected and subjected to virus titration to quantify infectious virus.

## Quantification of cell growth and virus decay kinetics

To estimate the growth kinetics of target cells, we used the following mathematical model:

$$\frac{dT(t)}{dt} = gT(t)\left(1 - \frac{T(t)}{K}\right),\tag{1}$$

where the variable $T(t)$ represents the number of uninfected target cells (cells/ml) at time $t$, and the parameters $g$ and $K$ indicate the growth rate and the carrying capacity of the target cells (cells/ml), respectively. Using the non-linear least square method, we fitted the model to the time-course growth data of cells (see Growth of HEK293-C34 cells and in S2A Fig) and estimated $g$ and $K$.

Furthermore, we estimated the clearance rate of infectious viruses, $c$, by a simple exponential decay model:

$$\frac{dV(t)}{dt} = -cV(t),\tag{2}$$

where $V(t)$ represents the amount of infectious virus ($TCID_{50}$/ml) in the culture medium at time $t$. Linear regressions yield $c$ from the time-course degradation data of infectious viruses (see Degradation rate of HiBiT-carrying SARS-CoV-2 and S2B Fig). The estimated parameter values are summarized in Table 2.

## Mathematical model for SARS-CoV-2 infection

We employed the following mathematical model for SARS-CoV-2 infection in cell culture considering the antiviral efficacy of RDV:

$$\frac{dT(t)}{dt} = gT(t)\left(1 - \frac{T(t) + E(t) + I(t)}{K}\right) - \beta T(t)V(t),\tag{3}$$

$$\frac{dE(t)}{dt} = gE(t)\left(1 - \frac{T(t) + E(t) + I(t)}{K}\right) + \beta T(t)V(t) - kE(t), \tag{4}$$

$$\frac{dI(t)}{dt} = kE(t) - \delta I(t), \tag{5}$$

$$\frac{dV(t)}{dt} = (1 - \varepsilon)\eta pI(t) - cV(t), \tag{6}$$

where $T(t)$, $E(t)$, and $I(t)$ are the numbers of uninfected target cells, eclipse phase cells, and virus-producing cells (cells/ml) at time $t$, respectively, and $V(t)$ is the amount of infectious virus ($TCID_{50}$/ml) at time $t$. The uninfected target and eclipse phase cells divide in logistic manner at rate $g$ and carrying capacity $K$. The target cells are infected by viruses at rate $\beta$, and the virus-infected cells stay in the eclipse phase during the period $1/k$. After this, they become virus-producing cells. The progeny viruses are produced by the virus-producing cells at rate $p$. The parameters $\delta$ and $c$ indicate the death rate of infected cells and the clearance rate of viruses, respectively. The inhibition rate of virus production by RDV is assumed to be $\varepsilon$. The fold-change of virus production rates of RDV-resistant viruses compared with WT virus are $\eta$ (i.e., $\eta = 1$ for WT virus).

### Data fitting and parameter estimation

The parameters $g$, $K$, and $c$ were independently estimated and fixed. A statistical model adopted in Bayesian inference assumed that measurement error followed a normal distribution with mean zero and constant variance (error variance). A gamma distribution was used as a prior distribution, and it inferred a distribution of error variance. As an output of MCMC computations, the posterior predictive parameter distribution represented parameter variability, and it inferred distributions of model parameters and initial values of variables. The estimated parameters and initial values are listed in Tables 2 and 3.

### Theoretical predictions and analyses of the effects of amino acid mutations

The WT NSP12 structure used as a reference was created based on the crystal structure (PDB ID: 6XEZ) [47]. From this crystal structure, only the NSP12 protein and 28 bases of the binding RNA (binding site) were extracted. The target mutant structures were constructed by substituting amino acids in the above WT structure. Based on the crystal structure (PDB ID: 7BV2) [48], the terminal nucleotides of each predicted structure were replaced with RDV. To neutralize the charge of these complex structures, counter ions were placed, and sufficient water molecules were placed around them. Each structure was stabilized by the energy minimization method and used as the initial structure for molecular dynamics simulations. The composite structure was thermally stabilized by raising the temperature from 0 K to 310 K (*in vivo* temperature, approximately 36.85˚C) over 500,000 steps with $\Delta t = 0.2$ fs. The structural changes during this temperature increase process were structurally sampled for each complex at every 1000 steps. These sampling structures were superimposed, and the RMSD for each protein was calculated. In addition, to observe the extent to which the structural properties differed between WT and mutants, we superimposed them in various substructures and calculated and compared the RMSD differences. These energy minimizations and molecular dynamics simulations were performed with the AMBER18 program package [49]. The

"AMBER99 [50]", "GAFF [51]" and "TIP3P [52]" force fields for the "proteins and nucleic acids", "RDV", and "water molecules" were employed, respectively.

## Animal experiments

Syrian hamsters (male, 4 weeks old) were purchased from Japan SLC. Baseline body weights were measured before infection. For the virus infection experiments, hamsters were anesthetized by intramuscular injection of a mixture of 0.15 mg/kg medetomidine hydrochloride (Domitor, Nippon Zenyaku Kogyo), 2.5 mg/kg butorphanol (Vetorphale, Meiji Seika Pharma) and 4.0 mg/kg alphaxaone (Alfaxan, Jurox). WT and R10/E796G/C799F viruses ($10^4$ TCID$_{50}$ in 100 μl) or saline (100 μl) was intranasally inoculated under anesthesia. Oral swabs were collected at the indicated timepoints and body weight was recorded daily by 7 dpi. The viral RNA load in the oral swabs was determined by RT-qPCR. Total RNA was extracted from the oral swabs using a PureLink RNA mini kit (Invitrogen). The RNA was used as the template for RT-qPCR performed in accordance with the manufacturer's protocol using the One Step Prime-Script III RT-qPCR Mix (Takara) and the following primers and probe: Forward, 5'-CAC ATT GGC ACC CGC AAT C-3'; Reverse, 5'-GAG GAA CGA GAA GAG GCT TG-3'; Probe, FAM-ACT TCC TCA GGG AAC AAC ATT GCC A-BHQ. These primers target the *nucleocapsid* gene of SARS-CoV-2. Fluorescent signals were acquired using a QuantStudio5 Real Time PCR System (Applied Biosystems).

## Statistical analysis

Results are indicated as the means ± standard deviations or standard errors. Statistical significances were determined by two-tailed Student's t test, the one-way ANOVA with Dunnett's test or the Kruskal–Wallis test with the two stage linear step-up procedure of Benjamini, Kreiger, and Yekutieli, which was performed using GraphPad Prism (Software ver. 9.2.0). Significantly different values are indicated by asterisks (*$p<0.05$ or ***$p<0.001$).

## Supporting information

**S1 Fig. RDV susceptibility of the mutants.** Cells were infected with HiBiT-carrying SARS-CoV-2 viruses in the presence or absence of RDV (0–1.0 μM final concentration) for 48 hours. Luciferase activities were measured and normalized to no RDV treatment. EC50 was calculated using the drc package (v3.0–1).
(PPTX)

**S2 Fig. Biological characterization of HEK293-C34 cells and SARS-CoV-2.** (A) Growth kinetics of HEK293-C34 cells. HEK293-C34 cells were counted for 48 hours after seeding. (B) Degradation rate of SARS-CoV-2. Virus titers were determined every 12 hours during incubation at 37°C.
(PPTX)

**S3 Fig. SARS-CoV-2 infection dynamics without and with RDV.** Solid curves and shadowed regions correspond to the best-fit solution and 95% posterior intervals, respectively, of Eqs (3–6) for the time-course dataset (black and blue dots). Top and bottom panels correspond to experiments without and with RDV treatment, respectively. All data for each strain were fitted simultaneously.
(PPTX)

**S4 Fig. RNA binding sites in NSP12 protein.** (A) Surface notation for NSP12 protein and van der Waals notation for bound RNA. (B and C) The RNA binding space is exposed by using

only the protein surface notation and cartoon notation.
(PPTX)

**S5 Fig. RNA binding sites in NSP12 protein.** The structure of SARS-CoV-2 replication-transcription complex indicated in the PDB ID:501 6XEZ (Whole-model, A) and the SARS-CoV-2 NSP12 (Small-model, B). (C) RMSD comparison of Whole-model and Small-model.
(PPTX)

**S1 Table. Viral genome sequencing of P10 with RDV and P10 without RDV viruses by MiSeq.**
(DOCX)

**S2 Table. Viral genome sequencing of single virus colonies.**
(DOCX)

**S3 Table. Primers to introduce mutations.**
(DOCX)

**S4 Table. Virus titers in the serial passages of SARS-CoV-2.**
(DOCX)

**S1 Acknowledgement. Membership of The Genotype to Phenotype Japan (G2P-Japan) Consortium.**
(DOCX)

## Acknowledgments

We thank M. Tomiyama for her secretarial work, M. Ishibashi and K. Toyoda for their technical assistance. We also thank Dr. J. Sakuragi in Kanagawa Prefectural Institute of Public Health and Dr. M. Shimojima at NIID for providing SARS-CoV-2 strain JPN/TY/WK-521 and Bio Matrix Research for providing anti-SARS-CoV-2 NP monoclonal antibodies.

We thank Gillian Campbell, PhD, from Edanz (https://www.jp.edanz.com/ac), for editing a draft of this manuscript.

## Author Contributions

**Conceptualization:** Shiho Torii, Kwang Su Kim, Jun Koseki, Teppei Shimamura, Shingo Iwami, Takasuke Fukuhara.

**Data curation:** Shiho Torii, Kwang Su Kim, Jun Koseki.

**Formal analysis:** Shiho Torii, Kwang Su Kim, Jun Koseki, Rigel Suzuki, Jumpei Ito.

**Funding acquisition:** Teppei Shimamura, Shingo Iwami, Takasuke Fukuhara.

**Investigation:** Shiho Torii, Kwang Su Kim, Jun Koseki, Rigel Suzuki, Shoya Iwanami, Yasuhisa Fujita, Yong Dam Jeong, Jumpei Ito, Hiroyuki Asakura, Mami Nagashima, Kenji Sadamasu, Kazuhisa Yoshimura, Yoshiharu Matsuura.

**Methodology:** Shiho Torii, Kwang Su Kim, Jun Koseki, Rigel Suzuki, Shoya Iwanami, Yasuhisa Fujita, Yong Dam Jeong, Jumpei Ito, Hiroyuki Asakura, Mami Nagashima, Kenji Sadamasu, Kazuhisa Yoshimura, Yoshiharu Matsuura, Teppei Shimamura, Shingo Iwami, Takasuke Fukuhara.

**Resources:** Shiho Torii, Rigel Suzuki.

**Software:** Kwang Su Kim, Jun Koseki, Shoya Iwanami, Teppei Shimamura, Shingo Iwami.

**Supervision:** Kei Sato, Yoshiharu Matsuura, Teppei Shimamura, Shingo Iwami, Takasuke Fukuhara.

**Validation:** Shiho Torii, Kwang Su Kim, Jun Koseki, Teppei Shimamura, Shingo Iwami, Takasuke Fukuhara.

**Visualization:** Shiho Torii, Kwang Su Kim, Jun Koseki.

**Writing – original draft:** Shiho Torii, Kwang Su Kim, Jun Koseki.

**Writing – review & editing:** Shiho Torii, Kwang Su Kim, Jun Koseki, Rigel Suzuki, Teppei Shimamura, Shingo Iwami, Takasuke Fukuhara.

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
