## [Decision Letter · Decision Letter 0]

6 Oct 2022

Dear dr Fukuhara,

Thank you very much for submitting your manuscript "Increased flexibility of the SARS-CoV-2 RNA-binding site causes resistance to remdesivir" for consideration at PLOS Pathogens. As with all papers reviewed by the journal, your manuscript was reviewed by members of the editorial board and by several independent reviewers. In light of the reviews (below this email), we would like to invite the resubmission of a significantly-revised version that takes into account the reviewers' comments.

Both reviewers felt that apart from the Sanger sequencing used, a sequencing method with higher depth of reads would have been better to reveal relevant mutations that could have been missed. In addition, special attention is asked with respect to figureS3 as one of the reviewers observed that data points in the graphs seem to overlap.

We cannot make any decision about publication until we have seen the revised manuscript and your response to the reviewers' comments. Your revised manuscript is also likely to be sent to reviewers for further evaluation.

Sincerely,

Bart L. Haagmans

Guest Editor

PLOS Pathogens

Ron Fouchier

Section Editor

PLOS Pathogens

Kasturi Haldar

Editor-in-Chief

PLOS Pathogens

orcid.org/0000-0001-5065-158X

Michael Malim

Editor-in-Chief

PLOS Pathogens

orcid.org/0000-0002-7699-2064

Both reviewers felt that apart from the Sanger sequencing used, a sequencing method with higher depth of reads would have been better to reveal relevant mutations that could have been missed. In addition, special attention is asked with respect to figureS3 as one of the reviewers observed that data points in the graphs seem to overlap.

Reviewer's Responses to Questions

**Part I - Summary**

Reviewer #1: Here, Torii, et al. examined mutations in the SARS-CoV-2 nsp12 protein conferring resistance to remdesivir, both those found through their own serial passaging and from those found through other studies. They created recombinant virus carrying those mutations and ran experiments showing increased titer and infection rates by these mutants relative to WT virus in the presence of remdesivir. In addition, they ran mathematical models to examine the effects of the mutations on viral production, and to show that these mutations increase molecular vibration around the RNA-binding site of nsp12.

While not the first in doing serial passaging of SARS-CoV-2 in the presence of remdesivir, this study found mutations that have not been found in other papers. The mathematical model for measuring viral production is a unique aspect of this paper, and molecular dynamics simulations is a step further than what is often done in similar studies that examine structural changes in mutated proteins. Incorporating mutations found in published studies was a nice idea.

The passaging experiment setup seems sound, as are the time course assays and the use of HiBiT. However, the main weakness of the paper comes from the use of Sanger sequencing and what seems to be a lack of replicates for the serial passaging.

Reviewer #2: This paper attempted to identify multiple Remdesivir resistant mutations withing the NSP12 gene of SARS-CoV-2 and examine mechanisms of this resistance. They achieved their aim by using a mixture of classical virology methods, reverse genetics to recover mutant viruses, mathematical modelling, and molecular dynamics analysis. Torii et al. identified novel mutations that have not been described previously. However, their discovery supports the hypothesis that mutations within the RNA-binding site of NSP12 aid the virus in resistance to Remdesivir. Also, different mutations arising could have something to do with different cell type used in previous studies. Observation of remdesivir resistant mutants showing lower infection rates and slower growth rates suggests that they are outcompeted by the wild type virus, and explains why such mutations and remdesivir resistance are not very common in patients. The manuscript overall is well written.

**Part II – Major Issues: Key Experiments Required for Acceptance**

Reviewer #1: Major points:

Sanger sequencing was used to discover the mutations that emerged from the serial passaging experiment, particularly those shown in Figure 1B. However, in recent years next-generation sequencing (NGS) methods have become much more prevalent in searching for mutations, so there could be less prevalent but relevant mutations that could have been missed.

Because of the limits of Sanger sequencing, how sure are the authors that all of the mutations found in the serial passaging can occur on the same individual viral genome, like the R10/C799F/E796G mutant that was constructed? Is there information on how many separate replicates or lineages were created during the passaging? Was there only one? Were viruses from passages other than P10 sequenced? What were the controls? Were titers of the passages measured? If there was more than one lineage of viruses that was passaged, it would establish replicability and show that these mutations primarily arise in vitro, since they are not present in the NextStrain database. Having data for controls would also indicate whether the non-nsp12 mutations in particular are random or not.

Figure 4 – In regard to the images of the models: while the authors point to where the active site of nsp12 is, there is no information where any of the mutated amino acids are located on the protein. At the very least, it would be good to include for each model in Fig. 4C. This would be especially important if any of the residues examined in this paper are near the active site, which could affect interactions between RDV and those amino acids. Even if they are not, pinpointing their location in structural motifs would also help corroborate the results found in the modeling. As it stands, it feels as if more information could be included in this figure.

Reviewer #2: I didn't see anywhere in the manuscript that the sequences of the resistant mutants have been submitted to appropriate database. These sequences should be made available to the scientific community.

**Part III – Minor Issues: Editorial and Data Presentation Modifications**

Reviewer #1: Minor points:

A justification for the concentrations of RDV used would be good to add along with the missing information mentioned in the previous section

Line 121 – Looking up the sequence of JPN/TY/WK-521 on GenBank led to a web page where it seemed like the record was removed, although the sequence was still there through a link on that page. Not necessarily an issue but if there is an updated record of the sequence a link to it would be appreciated, as well as information on the strain(s) used if available.

Line 124 – Comparing this with Figure 1A was confusing since Fig 1A shows that the passages started at 0.01 µM, and that it took for days for CPE to appear before supernatant was collected for passage 2. However, there doesn’t seem to be any period of 10 days mentioned in Fig 1A, so I assume this is referring to taking the initial stock (which is called P1 in the experiment) and treating it with 0.1 µM? It would be great if this were made clearer.

Lines 304-307 – This sentence is awkwardly constructed. Perhaps change the phrase “didn’t gain the pathogenicity nor fitness” to something like “had neither gained pathogenicity nor fitness” to better phrase the point. In general, contractions such as “didn’t” are avoided when writing in formal English.

Lines 458 and 466 – These equations should be labeled (1) and (2), which makes the subsequent equations numbers 3-6

Additional points to consider for your discussion:

Given that the active polymerase complex is not solely comprised of nsP12, but also includes nsP7 and nsP8, is there a chance that they plus nsP13 (which is also present in the original structure that was used for the modeling (PDB ID:501 6XEZ)) could have affected the results?

Since the F480L and V537L mutations were based off similar mutations in MHV, would you expect to see similar results when modeling the thermodynamic stability of MHV nsP12? There could be a difference in how F480L affects the protein between SARS-CoV-2 and MHV that explains the lack of conferral of resistance to RDV in the former.

Would a competitive fitness assay or an in vivo study with the mutants bolster the findings?

Equations for growth kinetics and viral production are helpful for modeling viral infections, but do polymerase mutations have that much of an effect on viral production? There’s an assumption that none of the other parts of the viral life cycle are affected in any way. HiBiT works for measuring production of mature virions but none of this measures polymerase activity directly.

While the paper in which the CPER method was established is appropriately cited, certain specifics such as the primers used to generate the amplicons and the full-length genome aren’t mentioned here at all

Reviewer #2: Authors used Sanger sequencing to determine the full-length sequence and identify mutations conferring resistance to Remdesivir. A sequencing method with higher depth of reads would have been better, as this would have provided insight into variability at the location of the mutations. Also, authors might have missed other interesting mutations. However, mutations that were identified, have been confirmed to contribute to resistance.

Line 204 and 573 – the text says that virus production was measured for 96h, but the graphs in figure 2 show timepoints only up to 72 hours.

Lines 219-222 – there is a discrepancy between the numbers mentioned in the text and the figure 3C. Mean antiviral effect for R10/E796G/C799F and E802D according to the figure is just below 40% and not below 30% as suggested in the text. Also values in the text don’t match values in Table 3. This needs to be checked and clarified.

Line 361 – typo: 10 uM of the supernatant couldn’t have been passaged; 10%? 10 ul?

Lines 403 – 407 – Method used to calculate TCID50 should be mentioned/cited.

Figure S3 – “virus producing cell” graphs for mutations E796G, C799F, R10/E796G/C799F, R10/C799F, E802D, D484Y, F480L/V557 look almost the same for the “without RDV” and the “with RDV”. The data points in these graphs seem to overlap. This needs to be double checked in case of a copy/paste error.

PLOS authors have the option to publish the peer review history of their article (what does this mean?). If published, this will include your full peer review and any attached files.

Reviewer #1: No

Reviewer #2: **Yes: **Agnieszka M. Szemiel
---

## [Editor Report · Decision Letter 1]

22 Feb 2023

Dear dr Fukuhara,

We are pleased to inform you that your manuscript 'Increased flexibility of the SARS-CoV-2 RNA-binding site causes resistance to remdesivir' has been provisionally accepted for publication in PLOS Pathogens.

Best regards,

Bart L. Haagmans

Guest Editor

PLOS Pathogens

Ron Fouchier

Section Editor

PLOS Pathogens

Kasturi Haldar

Editor-in-Chief

PLOS Pathogens

orcid.org/0000-0001-5065-158X

Michael Malim

Editor-in-Chief

PLOS Pathogens

orcid.org/0000-0002-7699-2064
---

## [Editor Report · Acceptance letter]

22 Mar 2023

Dear Dr. Fukuhara,

We are delighted to inform you that your manuscript, "Increased flexibility of the SARS-CoV-2 RNA-binding site causes resistance to remdesivir," has been formally accepted for publication in PLOS Pathogens.

Best regards,

Kasturi Haldar

Editor-in-Chief

PLOS Pathogens

orcid.org/0000-0001-5065-158X

Michael Malim

Editor-in-Chief

PLOS Pathogens

orcid.org/0000-0002-7699-2064